# An Off-the-Shelf Approach for the Production of Fc Fusion Proteins by Protein *Trans*-Splicing towards Generating a Lectibody In Vitro

**DOI:** 10.3390/ijms21114011

**Published:** 2020-06-03

**Authors:** Anniina Jaakkonen, Gerrit Volkmann, Hideo Iwaï

**Affiliations:** 1Institute of Biotechnology, University of Helsinki, FI-00014 Helsinki, Finland; anniina.jaakkonen@helsinki.fi (A.J.); gerrit.volkmann@gmx.de (G.V.); 2Present Address: Microbiology Unit, Finnish Food Authority, FI-00790 Helsinki, Finland

**Keywords:** protein *trans*-splicing, split intein, immunoglobulin G, lectin, protein ligation, Fc, scytovirin, segmental isotopic labeling, lectibody

## Abstract

Monoclonal antibodies, engineered antibodies, and antibody fragments have become important biological therapeutic platforms. The IgG format with bivalent binding sites has a modular structure with different biological roles, i.e., effector and binding functions, in different domains. We demonstrated the reconstruction of an IgG-like domain structure in vitro by protein ligation using protein *trans*-splicing. We produced various binding domains to replace the binding domain of IgG from *Escherichia coli* and the Fc domain of human IgG from *Brevibacillus choshinensis* as split-intein fusions. We showed that in vitro protein ligation could produce various Fc-fusions at the N-terminus in vitro from the independently produced domains from different organisms. We thus propose an off-the-shelf approach for the combinatorial production of Fc fusions in vitro with several distinct binding domains, particularly from naturally occurring binding domains. Antiviral lectins from algae are known to inhibit virus entry of HIV and SARS coronavirus. We demonstrated that a lectin could be fused with the Fc-domain in vitro by protein ligation, producing an IgG-like molecule as a “lectibody”. Such an Fc-fusion could be produced in vitro by this approach, which could be an attractive method for developing potential therapeutic agents against rapidly emerging infectious diseases like SARS coronavirus without any genetic fusion and expression optimization.

## 1. Introduction

Immunoglobulin G (IgG) is a multidomain protein that consists of four polypeptide chains. Two identical heavy chains and two identical light chains form a Y-shaped molecule connected by disulfide bonds (Figure 1a). While the heavy chain is composed of four structural domains called immunoglobulin domains (V_H_, C_H_1, C_H_2, and C_H_3), the light chain has two immunoglobulin domains (V_L_ and C_L_) (Figure 1a) [1]. Variable domains V_H_ and V_L_ constitute an antigen-binding region, bearing two binding sites per IgG. The enormous diversity of the variable domains plays an essential role in recognizing specific antigens, such as pathogens and foreign molecules in immunity [1]. Constant domains C_H_2 and C_H_3 of the two heavy chains construct the Fc domain, which mediates various effector functions such as antibody-dependent, cell-mediated cytotoxicity (ADCC) and the activation of complement pathways in immunity. Thus, domain structures of antibodies can be dissected based on their functional roles. Antibodies from camelids (like camel and llama) lack light chains, and are therefore termed as heavy-chain antibodies (Figure 1b) [2]. Antigen-binding regions of the heavy-chain antibodies contain only one immunoglobulin fold termed V_H_H. The V_H_H domain replaces the functional role of the variable domains V_H_ and V_L_ in conventional immunoglobulins [2]. The smaller size of V_H_H provides several advantages, such as simple production and engineering [2]. The V_H_H domain has been widely used as a standalone agent for biotechnological applications [3,4].

Monoclonal antibodies such as IgG have increasingly become indispensable biological therapeutic drugs for cancer and autoimmune diseases, with currently more than 70 antibody drugs on the market [6]. Because of their therapeutic success, antibodies represent the fastest growing sector in the pharmaceutical industry. Therapeutic antibodies typically comprise intact monoclonal IgG molecules and IgG derivatives [6]. Additionally, antibody fragments have been fused with various functional moieties, including enzymes, toxins, radionuclides, and viruses, for new therapeutic functions [3,4]. Since the therapeutic efficacy of ADCC by antibodies is mitigated by the fact that not all immune cells express antibody receptors, bispecific antibodies have been artificially engineered to simultaneously bind to two different types of antigens (Figure 1c) [7]. All IgGs, heavy-chain antibodies, and bispecific antibodies commonly contain an Fc domain with some sequence variation, while the binding sites are unique to individual formats and antigens. mAbs, Fc fusion proteins, and bispecific antibodies are usually produced from a single cell line for each BsAb, mAb, and Fc fusion by making stable cell lines to produce the genetically engineered proteins, despite the likely presence of the Fc domain among these proteins [8]. Whereas a binding domain such as V_H_H can be obtained by in vitro selection with directed evolution [9], the production of each new mAb from a mammalian cell line can be labor-intensive and time-consuming, especially for bispecific antibodies requiring four separate constructs (encoding each of the four polypeptide chains for the production of BsAbs) to be engineered [10].

We aimed to develop an efficient method to produce diverse N-terminal fusion proteins of the Fc domain in vitro from dissected domains by protein *trans*-splicing (Figure 1d,e). Protein splicing is an autocatalytic selfexcision process catalyzed by intein and concomitantly ligating the flanking protein sequences (exteins), catalyzed by an intein (Figure 2a) [11,12,13]. Canonical inteins contain a homing endonuclease (EN) domain. Inteins lacking EN domains are also found in nature and called mini-inteins (Figure 2b) [12]. Split inteins can catalyze bimolecular protein splicing, namely protein splicing in *trans* (Figure 2b) [12]. Since protein *trans*-splicing (PTS) was established as a protein-ligation tool of foreign protein sequences with a peptide bond, it has become an essential protein engineering tool (Figure 2d) [14,15]. The split fragments of an intein are termed N-intein (Int_N_) and C-intein (Int_C_) for N-terminal and C-terminal fragments, respectively [13,14]. Many artificially and naturally split inteins have been developed and characterized for in vitro protein ligation (Figure 2c) [14,16,17]. In vitro protein ligation of Fc domain with diverse binding domains could circumvent the fusion of different genes on the DNA level with an identical Fc gene, facilitating in vitro production of Fc fusions with various binding capabilities using an off-the-shelf approach (Figure 1e and Figure 3) [18,19].

In this study, we prepared split-intein fusions with the human Fc domain produced in *Brevibacillus choshinensis*, a gram-positive bacterium. *Brevibacillus choshinensis* has an excellent ability to produce many exogenous proteins extracellularly [23,24]. We tested in vitro protein ligation between the Fc domain and different model proteins, including the cyanobacterial lectin scytovirin tin (SVN). SVN has antiviral activity against a variety of human pathogens, including the HIV-1 and SARS viruses, by binding to glycosylated viral surface proteins [25,26,27,28]. The lectin-Fc fusion might thus function as a carbohydrate-targeting antibody, namely ‘lectibody’ (Figure 3) [29].

## 2. Results

### 2.1. Strategy to Produce Fc Fusions In Vitro

We dissected the IgG format at two different sites within the hinge region (Figure 1d and Figure 4a). The hinge region can be divided into the upper, core, and lower hinges [1,30]. The Fc domain with and without the core region was fused with the C-terminal fragment (Int_C_) of the naturally split DnaE intein from *Synechocystis* species, strain PCC6803 (*Ssp*DnaE_C16_) (Figure 4b). We chose *Npu*/*Ssp*DnaE chimeric intein for protein ligation because it has been used successfully in the ligation of many protein domains in vitro, and has a higher tolerance of amino acids at the +2 position [21]. Naturally split *Ssp*DnaE and *Npu*DnaE inteins have a 36-residue Int_C_ fragment (Figure 2c) [21]. Therefore, we termed the split-site of these inteins as C35 (excluding the start codon Met residue) [12]. However, we selected C15 as the intein split site, and used the engineered pair of *Npu*DnaE_ΔC16_/*Ssp*DnaE_C16_, because this pair has worked better for some model proteins than the naturally split C35 pair of split inteins (Figure 2c) [22]. We hypothesized that the shorter 15-residue Int_C_ would disturb less the folding of the Fc domain. Three different Int_C_-Fc fusion proteins (Int_C_-Fc, Int_C_+1S-Fc, and Int_C_-CPPC-Fc) were constructed in this study (Figure 4b).

### 2.2. Cloning and Protein Expression of Int_C_-Fc in Brevibacillus choshinensis

The Fc domain without the hinge-region has been produced in *Escherichia coli* [31,32,33]. We decided to use the commercially available *Brevibacillus* expression system which is suitable for the secretory production of heterologous proteins up to 3.7 g/L [34,35]. Following the protocol of the commercial *Brevibacillus* expression system (TakaraBio), we cloned the gene of Int_C_-Fc fusion protein with or without the core hinge into pNY326 and pNCM02 with the signal peptide of HWP from the kit (Figure 4b). *Brevibacillus choshinensis* has a distinctive cell surface structure containing one surface protein layer formed by HPD31 cell wall protein (HWP) under the regulation of five tandem promoters, named P1–P5. While pNY326 use P5 promoter, pNCM02 is a high-copy-number plasmid harboring a strong promoter, P2, and a modified Sec signal sequence for efficient secretory expression in *B. choshinensis* [35].

We then compared the protein expression of the Fc fusions in TM and 2SY media (Figure 4c and Appendix A). We found that the cells harboring the pNCM02-backbone vector secreted Int_C_-Fc fusion with the core hinge more quickly in 2SY than the cells bearing pNY326 backbone vector in TM and SY media (Appendix A).

### 2.3. Production of IntC-Fc for Protein Trans-Splicing

For the production of Int_C_-Fc fusions for protein ligation, we chose three different constructs Int_C_-Fc (AJNCM18P), Int_C_+1S-Fc (AJNCM20A), and Int_C_-CPPC-Fc (AJNCM21A) (Figure 4). We constructed Int_C_-Fc with or without the core hinge. Additionally, we created a variant replacing Cys+1 with Ser+1 for the active site of the intein at the so-called +1 position [21]. We first optimized the culture media and analyzed the time-course of the expression using AJNCM18P (Int_C_-Fc without the core hinge) as a model protein. The presence of the Int_C_-Fc was detected in both TM and 2SY media after three days, and no significant increase was observed after four days (Figure 4c). Based on these results, three different Int_C_-Fc fusion proteins were expressed and secreted under the P2 promoter in a 4-mL scale using TM medium and compared after four days (Figure 4d). All three fusion proteins were successfully secreted into the culture medium. Whereas the control protein HWP with the molecular weight of 118 kDa accumulated mostly in the supernatant fraction, Int_C_-Fc and Int_C_+1S-Fc were also detected in the pellet fraction, suggesting incomplete secretion of the fusion proteins or incorrect folding of the protein. In contrast, Int_C_-CPPC-Fc was mainly present in the supernatant fraction.

### 2.4. Purification of Int_C_-Fc Fusions

We purified Int_C_-Fc fusions using an N-terminal hexahistidine (His-tag) incorporated in Int_C_ by Immobilized Metal Chelate Affinity Chromatography (IMAC) (Figure 4b and Figure 5). The yields were 27 mg and 15 mg per liter of the culture medium for Int_C_+1S-Fc and Int_C_-CPPC-Fc, respectively. These yields were calculated from pooled 4-mL cultures after three days of protein expression. It is noteworthy that we were unable to scale up to over 50 mL, abolishing the protein secretion of interest at a larger scale. After the first IMAC purification, we tested the binding of Int_C_-Fc fusions to the *Staphylococcal* protein A column by affinity chromatography. All three Int_C_-Fc fusions bound to a protein A sepharose column and could be eluted by acid elution as used for IgG purification, indicating that the overexpressed Fc domain is capable of binding to the protein A column, despite the N-terminal fusion of Int_C_; 35–49% of the total protein could be recovered after the protein A affinity chromatography and subsequent dialysis. The molecular mass of Int_C_-CPPC-Fc produced from *B. choshinensis* was confirmed (Appendix A).

### 2.5. Model Proteins for Protein Ligation with Int_C_-Fc

To test the protein ligation of the purified Int_C_-Fc fusions, we prepared the B1 domain of IgG binding protein G (GB1) and yeast SMT3 protein (SUMO domain) as the N-terminal target proteins. Both GB1 and SUMO domains were fused with an N-terminal His-tag and with the N-terminal split fragment (Int_N_) of *Npu*DnaE_ΔC16_ intein (*Npu*DnaE_N123_) for PTS. Both model proteins were previously prepared and tested for protein ligation with other model proteins [22]. H_6_-GB1-Int_N_ (*Npu*DnaE_ΔC16_) and H_6_-SUMO-Int_N_ (*Npu*DnaE_ΔC16_) were independently produced in *E.coli* and purified by IMAC using the N-terminal His-tag (Figure 5b).

### 2.6. Protein Ligation of Int_C_-CPPC-Fc by Protein Trans-Splicing

The individually purified N-terminal precursor proteins (H_6_-GB1-Int_N_ and H_6_-SUMO-Int_N_) and three Fc fusion proteins (Int_C_-Fc, Int_C_+1S-Fc, and Int_C_-CPPC-Fc) were tested for in vitro protein ligation (Figure 5). The incubation of a reaction mixture of Int_C_-CPPC-Fc and H_6_-GB1-Int_N_ did not result in any ligated product (H_6_-GB1-CPPC-Fc) (Figure 5c). However, the same reaction mixture in the presence of a reducing agent, 0.5 mM TCEP (tris(2-carboxyethyl)phosphine), produced the ligated product H_6_-GB1-CPPC-Fc, together with the spliced-out intein fragment. Despite the slow reaction, i.e., requiring over 20 h, the in vitro ligation of Int_C_-CPPC-Fc with GB1 was possible under reducing conditions (Figure 5c). We also tested in vitro ligation using 5 mM DTT instead of TCEP, yielding a slightly faster reaction (Figure 5d). The requirement of a reducing agent suggests that disulfide bonds in the core hinge region might hinder the PTS reaction. Indeed, we observed the dimer formation of the Fc domain without a reducing agent, which could be prevented by removing the core hinge and Cys at the +1 position of Int_C_ (Appendix A). Next, we tested the SUMO domain as another model protein for in vitro protein ligation. Similar to GB1, we could produce H_6_-SUMO-CPPC-Fc in the presence of 0.5 mM TCEP (Figure 5e–g). In the case of H_6_-SUMO, we observed some premature cleavage of H_6_-SUMO-Int_N_, as reported previously [22]. Since two precursor proteins, i.e., H_6_-SUMO-Int_N_ and Int_C_-CPPC-Fc, have the same migration in SDS-PAGE, we could not estimate the reaction yield precisely (Figure 5e). However, the ligation with the SUMO domain accumulated the ligated product more than the ligation with GB1. We believe that this was because GB1 has an affinity to the Fc domain, thereby limiting its access to Int_C_-Fc for PTS.

### 2.7. The Influence of the Core Hinge on Protein Trans-Splicing

The core hinge in Int_C_-CPPC-Fc is more rigid due to two disulfide bonds, and could therefore complicate the folding of the Fc domain during protein production [30]. To evaluate the effect of these disulfide bridges on PTS, we tested in vitro protein ligation of Int_C_-Fc and Int_C_+1S-Fc, which lack the core hinge (Figure 5f,g). Int_C_+1S-Fc is expected to produce Fc fusion without any cysteine in the hinge region, while both Int_C_-CPPC-Fc and Int_C_-Fc result in an additional “CFN” sequence in the hinge region, which could form an interchain disulfide bond (Figure 4). The reaction mixture of H_6_-SUMO-Int_N_ and Int_C_-Fc produced the ligated product H_6_-SUMO-Fc only in the presence of 0.5 mM TCEP, similar to Int_C_-CPPC-Fc (Figure 5e,f). Premature cleavage of H_6_-SUMO-Int_N_ was observed both with Int_C_-Fc and Int_C_-CPPC-Fc, whereas the reaction of H_6_-SUMO-Int_N_ and Int_C_+1S-Fc did not show any premature cleavage of H_6_-SUMO-Int_N_. This observation is presumably because the hydroxyl group of serine is less nucleophilic than the thiol group of cysteine. Despite the slow reaction, the reaction mixture of H_6_-SUMO-Int_N_ and Int_C_+1S-Fc produced the ligated product H_6_-SUMO-S-Fc fusion. We expected that H_6_-SUMO-Int_N_ and Int_C_+1S-Fc would produce the ligated product without any reducing agent, because the mutation of Cys to Ser was introduced at the active site of the +1 position (Appendix A). However, Int_C_+1S-Fc still required a reducing agent, suggesting that other cysteine residues in the intein, such as the first residue of *Npu*DnaE intein, need reducing conditions.

### 2.8. In Vitro Production of the Lectin Fusion Protein by Protein Trans-Splicing

Encouraged by the results from the model proteins, we attempted to create ‘lectibody’ by replacing the antigen-binding domains of IgG with a small lectin by PTS (Figure 3 and Figure 6). Lectins are a group of proteins that recognize and bind carbohydrates with high specificity. Since there are characteristic carbohydrates, for example, on the surfaces of pathogens and cancer cells, lectins are promising candidates as neutralizing and drug-targeting agents [26,35]. For example, cyanobacterial lectin scytovirin (SVN) is an antiviral lectin from *Scytonema varium* that binds to high mannose oligosaccharides on the surface of the human immunodeficiency virus (HIV) [27]. Bivalent structures of the IgG format could increase the binding of lectin when it is fused with Fc. Additionally, the Fc domain could improve the in vivo stability of a small lectin protein.

We produced SVN as a fusion protein with thioredoxin (TRX) as previously reported, but additionally fused it with Int_N_ (*Npu*DnaE_ΔC16_) [27]. The construct TRX-H_6_-SVN-Int_N_ was expressed in *E. coli* under T7 promoter and purified with a His-tag by IMAC (Figure 6a,b). We expressed and purified the model protein GB1 with Int_C,_ as previously reported [22,36]. The reaction mixture of TRX-H_6_-SVN-Int_N_ and Int_C_-GB1 in the presence of 0.5 mM TCEP produced the ligated product TRX-H_6_-SVN-GB1 after 8 h (Figure 6b). The ligated product was immediately subjected to proteolytic digestion by enterokinase (EK) using an EK site engineered between the His-tag and SVN. The proteolysis of TRX-H_6_-SVN-GB1 resulted in two proteins, TRX-H_6_ and SVN-GB1 (Figure 6a,b). After digestion with EK, we removed the His-tagged TRX-H_6_ by affinity chromatography using IgG sepharose (Figure 6b, lane 8). SVN contains ten cysteines which need to form five disulfide bonds correctly, despite the presence of free cysteines of Int_N_. These disulfide bonds facilitate SVN folding to the native conformation [27,28]. Since we had to use reducing conditions for PTS, we prepared segmentally isotope-labeled SVN-GB1 for the structural assessment of SVN. We used ^15^N-labeled TRX-H_6_-SVN-Int_N_ produced in ^15^N-labeled medium and unlabeled Int_C_-GB1, resulting in segmentally labeled [^15^N]-SVN-[^14^N]-GB1 (Figure 6c) [36,37]. The molecular weight of the ligated [^15^N]-SVN-[14N]-GB1 was also confirmed by mass-spectrometry (Appendix A). The [^1^H, ^15^N]-HSQC spectra of SVN with invisible GB1 tag did not show well-dispersed peaks typical for the well-folded globular domain, indicating that SVN was incorrectly folded or unfolded. The fusion with Int_N_ or the ligation in reducing conditions likely affected the correct folding of SVN. We assume that the ligated SVN-GB1 prepared in this study was biologically inactive.

### 2.9. In vitro Production of a Lectibody by Protein Trans-Splicing

Despite the analysis of SVN-GB1, we tested Int_C_-CPPC-Fc and TRX-H_6_-SVN-Int_N_ for protein ligation in the presence of 0.5 mM TCEP (Figure 6d,e). As observed with SUMO-Int_N_, cleavage reactions were apparent. The ligated product TRX-H_6_- SVN-CPPC-Fc accumulated after 3 h of reaction (Figure 6e). This observation confirmed that it was feasible to produce ’lectibody’ in vitro by ligating the Fc domain and a lectin using PTS, even though SVN bearing five disulfide bonds is not ideal for PTS, which requires reducing conditions. Several algal lectins contain no or few disulfide bridges such as griffithsin, which could be better candidates for ‘lectibody’ production by PTS [38].

## 3. Discussion

Inspired by heavy-chain antibodies from camelids composed of only two heavy chains, we decided to create Fc fusion proteins in vitro at the N-terminus of the Fc region. In heavy-chain antibodies, 50-kDa antigen-binding domains of IgG, which consist of V_H_, V_L_, C_L,_ and C_H_1 domains, are replaced with a smaller 12-kDa variable domain (V_H_H). We hypothesized that carbohydrate-binding proteins like lectins could be able to replace the role of the variable domain of V_H_H as a binding domain in IgG format when fused at the N-terminus of Fc domain (Figure 3 and Figure 6d). Such Fc fusion proteins might function like an antibody, i.e., ‘lectibody’, with a bivalent binding capability [29]. Previously, a lectibody was engineered as a genetic fusion of a lectin and Fc domain, and produced in plants [29]. The production of full-length antibodies or antibody fragments in large quantities can be challenging in bacterial overexpression systems because the disulfide bonds need to form correctly [39]. Mammalian expression systems have been established for the large-scale production of monoclonal antibodies, but establishing a cell-line for each new monoclonal antibody or a fusion protein requires time-consuming optimizations [40]. Additionally, protein production in mammalian cells can be more expensive and slower compared with bacterial expression systems [39]. Heterogenous expression using mammalian cells could also be more problematic than bacterial expression for engineered foreign proteins from different organisms, such as from plants and bacteria, even in small quantities. Therefore, we developed an in vitro protein ligation approach, which provides an alternative solution for the production of Fc fusions. The in vitro protein ligation approach enabled us to produce various fusion proteins using off-the-shelf domains that were preproduced in different suitable host organisms for each domain. This off-the-shelf protein ligation approach avoids time-consuming steps to optimize the expression and purification for each fusion protein. Once individual domains have been prepared and stored, only the protein ligation step needs optimization.

In this study, we demonstrated the feasibility of producing N-terminal fusion proteins of the Fc domain in vitro by PTS. PTS successfully ligated the Fc domain made in *Brevibacillus* and other domains produced in *E. coli*. The method worked for the Fc domain, both with and without the core hinge region, although the reaction required reducing conditions. C-terminal fusions of a full-length IgG and Fc domains by PTS have been demonstrated previously, resulting in functional IgGs with a high ligation efficiency (80%) under reducing reaction conditions [41,42]. Reducing conditions had to be employed in previous reports, as well as in this study. The reducing conditions affected neither antibody affinity nor integrity by the protein ligation using PTS [41,42]. This observation indicated that Fc-fusion proteins generated by PTS could be further exploited for various in vitro products of Fc fusions, including bispecific antibodies, if two orthogonal split inteins could be used [42,43]. We successfully demonstrated the production of an Fc fusion of SVN, which is a model lectin bearing five disulfide bridges, and the N-terminal split intein in *E. coli*, as previously reported [27,28]. However, the SVN domain in the ligated product by PTS appeared to be unfolded or misfolded, as revealed by NMR analysis using segmental isotopic labeling of SVN [36,37]. We believe that the folding problem of SVN was due to the reducing condition used in the protein ligation reaction by PTS, and that the ligated product was functionally inactive. We chose SVN from the cyanobacterium *Scytonema varium* as a model system. SVN binds to various viruses, including HIV glycoproteins, SARS coronavirus, and Ebola virus [28,38,44]. It might be possible to use the fusion protein of antiviral lectins as a therapeutic anti-HIV or anti-SARS drugs as a lectibody. In our study, however, SVN bearing five disulfide bonds turned out to be a nonideal candidate for in vitro protein ligation by PTS, at least with split DnaE inteins that require reducing conditions for PTS. Many other lectins contain no disulfide bridges, such as griffithsin from red algae, which also binds to gp120 and gp41 of HIV with high affinity, as well as to SARS coronavirus [38,45,46]. These lectins bearing no disulfide bonds could be more easily developed as lectibodies with therapeutic potential when the intein-Fc fusions are prepared from mammalian cells for the effector functions of IgG, such as ADCC. As the Fc domain could improve the pharmacokinetic properties, the Fc domain may be a good fusion partner for other binding domains [29].

We used the chimeric *Npu/Ssp*DnaE split intein with a cysteine residue at the +1 position, which is required for protein splicing reactions and remains in the ligated product. We replaced the cysteine with a serine residue at the +1 position in one of the Int_C_-Fc precursors (Int_C_+1S-Fc). This construct, however, still requires reducing conditions for PTS reaction, despite the lack of cysteine in both the intein and hinge regions. This result suggests that other cysteines, such as the first residue of the intein, which is responsible for the first step of the splicing reaction, still require reducing conditions. However, the produced Fc domain retained the capability of protein A binding, indicating the proper three-dimensional structure of the Fc domain by preserving the protein A binding surface. The structure was preserved presumably because two buried intramolecular disulfide bridges in Ig folds (C_H_2 and C_H_3) were not influenced by a reducing agent such as 0.5 mM TCEP (Appendix A). In the case of the Fc domain with core hinge, the two disulfide-bridges (CPPC) in the core hinge probably have to be reduced to avoid steric hindrance with the Int_N_ fragment. Several inteins in nature contain no cysteine residue within their sequence, including the +1 position [47]. These cysteine-free inteins might be alternatively exploited for PTS.

## 4. Materials and Methods

### 4.1. Bacterial Strains and Growth Conditions

*B. choshinensis* strain HPD31-SP3 (Takara Bio, Shiga, Japan), and *E. coli* Rosetta-gami 2 (DE3) pLacI (Merck, Kenilworth, NJ, US) and T7 Express (New England Biolabs, Ipswich, MA, US) were used for protein expression. MT medium, MT supplemented with 1.6 mM MgCl_2,_ and 2SY media were used for culturing *B. choshinensis* [35]. *E. coli* was cultured in Luria-Bertani (LB) medium. *B. choshinensis* was transformed by electroporation with a protocol adjusted from the previous report [48].

### 4.2. Construction of Plasmids

For the N-terminal target proteins, an *E.coli* expression vector for H_6_-SUMO-Int_N_ was constructed by transferring the gene of Int_N_ from pHYDuet36 [22] into pHYRSF53 vector [49] (addgene #64696) using *Bam*HI and *Hind*III restriction sites, resulting in pHYRSF53-36. The gene encoding TRX-H_6_-SVN was amplified from pET(SVN) by PCR with two oligonucleotides, HK901: 5′-TACACCATGGGAAGCGATAAAATTATTCAC and HK900: 5′-TAGGATCCAC TACCACCTCCCGCAGCCGC GTGACCCGC, followed by cloning into pHYRSF53-36 between *Nco*I and *Bam*HI sites, resulting in pAJRSF25A bearing the gene of TRX-H_6_-SVN-Int_N_ (*Npu*DnaE_ΔC15_).

Expression vectors of Int_C_-Fc fusions in *B. choshinensis* were cloned as follows. The gene encoding the DnaE C-intein (Int_C_) from *Synechocystis* sp. (*Ssp*DnaE_C16_) was constructed as described previously and transferred into the pBAD vector, resulting in pSABAD7PG [25]. The gene of Fc domain bearing C_H_2 and C_H_3from human immunoglobulin G1 was amplified from the L1-Fc vector (addgene #15124) and cloned into pSABAD7PG, resulting in pGVBAD049. The N-terminal hexahistidine tag followed by a thrombin cleavage site introduced by cloning the gene into pRSF vector with *Pst*I site introduced by PCR using two oligonucleotides of GV065: 5′- CAGGTAGCCATCATCATCATCATCACAGCAGC and GV066: 5′-CAGTATATCTCCTTATTAAAGTTAAACAAAATTATTTCTACAGG, resulting in pGVRSF55-049#2. The lower hinge sequence of ‘PAPELLGG’ was introduced by PCR assembly using the four oligonucleotides GV065, AJ001: 5′-CCAGCACCTGAACTCCTAGGGGGACCGTCAGTCTTCCTC, J002: 5′-CCCTAGGAGTTCAGGTGCTGGGTTAAAACAGTTGGCGGCGATAG, and HK548: 5′-TACAAGCTTATTTACCCGGAGACAGGG, and cloned into pGVRSF55-049#2 and pNCM02 vector using *Pst*I-*Hind*III restriction sites, resulting in pAJRSF18A and pAJNCM18P, respectively for the expression of Int_C_-Fc. For the production of Int_C_+1S-Fc in *B. choshinensis*, the vector pAJNCM18P was mutated by inverse PCR using a pair of the two primers HK866: 5′- GGTGCTATCGCCGCGAATTCTTTTAACCCAGCACCTG and HK867: 5′-CAGGTGCTGGGTTAAAAGAATTCGCGGCGATAGCACC, resulting in pAJNCM20A. For Int_C_-CPPC-Fc, the core hinge ‘CPPC’ was introduced into pAJRSF18P by inverse PCR using the two primers HK881: 5′-TTGCCCACCGTGCCCAGCACCTGAGCTCCTAGGGGGAC and HK882: 5′-GGTGCTGGGCACGGTGGGSAATTAAAACAGTTGGCGGCG, resulting in pAJRSF21B. The gene corresponding to Int_C_-CPPC-Fc was transferred from pAJRSF21B to pNCM02 or pNY326, resulting in pAJNCM21A and pAJNY21F vectors with two different promoters. The pAJNCM18P, pAJNCM20A, pAJNCM21A, pAJNY21F, pHYRSF53-36, and pAJRSF25A will be available from Addgene (www.addgene.org) with addgene #153166, #153167, #153168, #153169, #153170, and #153171, respectively.

### 4.3. Expression and Purification of Fc Fusions from B. choshinensis

The plasmids pAJNCM18P, pAJNCM20A, and pAJNCM21A were transformed into *B. choshinensis* for protein expression and incubated in 4 mL of TM media at 30 °C, 200 min^−1^ overnight. The precultures were then inoculated by a 100-fold dilution and incubated in 4 mL 2SY medium at 30 °C. After 3–4 days, the culture supernatants were collected and subjected to SDS-PAGE analysis of the protein incubation. The culture supernatants were collected, centrifuged (42,740× *g*, 50 min, 4 °C), and loaded onto a 5-mL HisTrap FF column (GE Healthcare, Chicago, IL, USA). Unbound contaminants were washed off with 35 mM imidazole, and bound proteins were eluted with a 15- or 20-mL linear gradient of imidazole (35 mM–250 mM) in 50 mM sodium phosphate, 300 mM NaCl, pH 8.0. Fractions were collected and monitored by measuring absorbance at 280 nm (A_280_), and their purity was confirmed by SDS-PAGE. The fractions containing target proteins were pooled and dialyzed overnight at 4 °C against 20 mM sodium phosphate, pH 6.5 (protein A binding buffer) using 6–8 kDa molecular weight cut-off (MWCO) membrane. The dialyzed samples were concentrated using a centrifugal filter device (Vivaspin 20, 3 kDa MWCO, GE Healthcare) and clarified by centrifugation (20,817× *g*, 15 min, 4 °C) prior to loading on a 1-mL HiTrap Protein A HP column. After washing, the bound proteins were eluted with a 2-mL linear gradient of 100 mM sodium citrate, pH 3.5 (0–100%). Fractions were collected and monitored by measuring A_280_ and by SDS-PAGE examination. The fractions containing the target proteins were immediately neutralized with 20% (*v*/*v*) 1 M Tris, pH 9.5, pooled, and dialyzed overnight at 4 °C against 10 mM Tris, 500 mM NaCl, 1 mM EDTA, pH 7 (ligation buffer). When necessary, the dialyzed samples were concentrated using a centrifugal filter device (Microcon YM-10, 10kDa MWCO, Millipore, Temecula, CA, US).

### 4.4. Expression and Purification of the N-Terminal Model Domains

Expression vectors pHYDuet36 [22], pHYRSF53-36, pAJRSF25A, and pSZRS7PG [22] were transformed into *E. coli* Rosetta-gami 2(DE3)pLacI. A single transformant was grown overnight in 50 mL LB medium, at 30–37 °C, 220 min^−1^. The overnight culture was diluted into 2 L of LB medium in three shake flasks and cultured at 37 °C, 220 min^−1^. When OD_600_ reached 0.5–0.6, protein expression was induced with IPTG in the final concentration of 40 μg/mL. After 4-h protein expression, cells were harvested by centrifugation (8,980× *g*, 10 min, 4 °C) and resuspended into 50 mM sodium phosphate, 300 mM NaCl, pH 8.0. The cell suspension was flash-frozen with liquid nitrogen and stored at −70 °C. After thawing, cells were lysed by sonication. Cell debris was removed by centrifugation (42,740× *g*, 50 min, 4 °C), and the expressed proteins were further purified by polyhistidine tag IMAC with a linear gradient of 50–250 mM imidazole. Elution from the HisTrap FF column was dialyzed overnight at 4 °C against 10–20 mM Tris, 500 mM NaCl, 0–1 mM EDTA, pH 7–8 using 6–8 kDa MWCO membrane.

### 4.5. Protein Trans-Splicing Assays

For ligation reactions, equal amounts of the two precursor proteins were mixed at a final concentration of 10–18 µM. The reactions were conducted using a bench-top shaker (Thermomixer Comfort R, Eppendorf) at 25 °C with mild shaking (350 rpm) in either reducing or nonreducing conditions. For reducing conditions, 0.5 mM TCEP and 5 mM DTT were tested. The reactions were monitored periodically after mixing by reducing SDS-PAGE. Each sample for SDS-PAGE was mixed 1:1 with 2× SDS sample buffer containing 200 mM DTT and heated at 95 °C for 5 min to stop the reaction. The samples were either loaded on a gel immediately or stored overnight at −20 °C.

SDS-PAGE and protein A affinity chromatography were used to analyze the effect of protein *trans*-splicing reaction conditions on intramolecular disulfide bonds. Proteins AJNCM18P, AJNCM20A, and AJNCM21A at a concentration of 15 µM were incubated in the presence of 0.5 mM TCEP or 5 mM DTT at 25 °C, 350 rpm (Thermomixer Comfort R, Eppendorf, Hamburg, Germany). Control reactions were incubated without a reducing agent. After 3 h, samples were mixed with the SDS sample buffer and analyzed on a gel. AJNCM21A at a concentration of 14 µM was incubated in the presence of 0.5 mM TCEP for 1.5 h and stored overnight at 4 °C. The buffer was exchanged for a protein A binding buffer in a centrifugal filter device (Amicon Ultra-4, 3-kDa MWCO, Millipore, Temecula, USA), and the sample was loaded onto the protein A column.

## 5. Conclusions

We demonstrated an off-the-shelf approach to producing Fc fusion proteins at their N-terminus by in vitro protein ligation using PTS. Independently prepared N-terminal domains made in *E. coli* were ligated in vitro with the C-terminal Fc domain produced in *Brevibacillus choshinensis*. The in vitro product could constitute an IgG-like Y-shape structure with bivalent binding sites. In nature, there are many natural bioactive proteins, peptides, and domains like lectins that bind to various receptors and ligands but which have not been widely used as a binding domain in IgG-like format. The production of fusion proteins with such bioactive domains in vitro using an off-the-shelf approach paves a new way for the rapid production of IgG-like molecules with known therapeutic potential without any time-consuming genetic engineering and optimization. The off-the-shelf approach using premade domains could be advantageous for drug development against rapidly emerging infectious diseases.

## Figures and Tables

**Figure 1 ijms-21-04011-f001:**
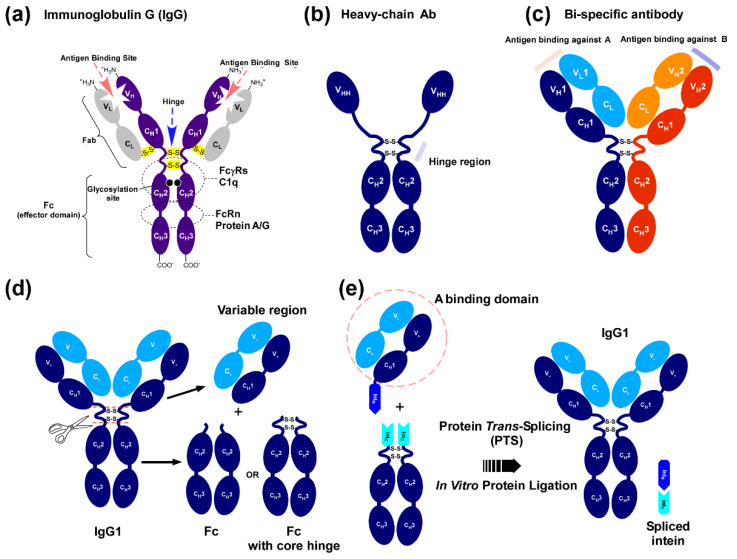
Domain structures of immunoglobulin G (IgG) and IgG variants. (**a**) Domains in IgG with their functional roles. (**b**) Domain structure of a heavy-chain antibody. (**c**) Domain structure of an engineered bispecific antibody showing each unique chain highlighted in different colors. Dissection of immunoglobulin G (IgG). (**d**) IgG can be split into the Fab region, which contains an antigen-binding site, and the effector domain Fc at two positions in the hinge region. (**e**) Reconstitution of IgG-like molecules by protein ligation using protein *trans*-splicing (PTS) with a split intein. The figure was adapted from [5].

**Figure 2 ijms-21-04011-f002:**
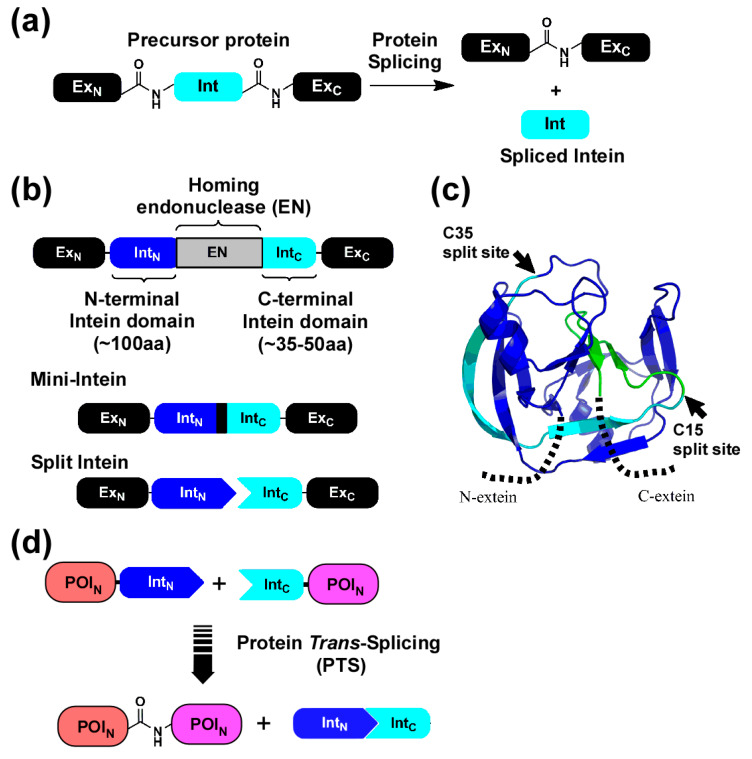
Protein splicing in *cis* and *trans.* (**a**) Protein splicing in *cis*. An internal protein (Intein, Int) is spliced out from a precursor protein, concomitantly ligating the two flanking proteins (N-extein and C-extein). Ex_N_ and Ex_C_ indicate N-extein and C-extein, respectively. (**b**) Canonical intein with homing endonuclease domain (EN), Mini-intein without EN, and split intein. (**c**) A cartoon model of *Npu*DnaE intein [20,21]. The natural split-site (C35) and engineered split-site (C15) are marked with arrows [22]. The N-terminal split fragment of the natural split intein is colored in blue. Int_N_ used in this study is colored in blue or cyan. The region corresponding to the Int_C_ fragment used in this study is highlighted in green. (**d**) Protein ligation by PTS using a split intein. PTS ligates the N-terminal fragment of a protein of interest (POI_N_) fused with the N-terminal split intein (Int_N_) and the C-terminal fragment of POI (POI_C_) fused with the C-terminal split intein (Int_C_). PTS results in ligated POI_N_-POI_C_ and excised Int_N_ and Int_C_. The figures were adapted from [5].

**Figure 3 ijms-21-04011-f003:**
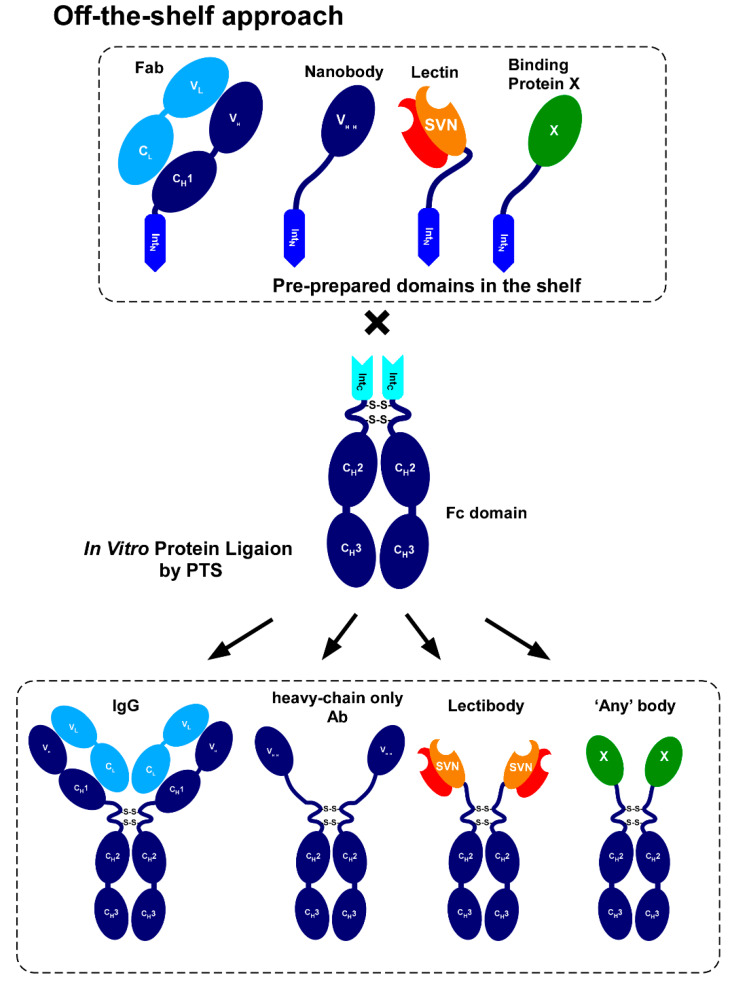
Off-the-shelf approach for in vitro production of Fc fusion by protein ligation. Various binding domains (protein of interest, POI), such as lectins and V_H_H, fused with Int_N_ are separately prepared. The Fc domain fused with Int_C_ is also individually made. The N-terminal POI and C-terminal Fc domain are ligated in vitro by PTS, concomitantly splicing out the two intein fragments Int_N_ and Int_C_. The ligated Fc fusion is a bivalent Y-shape molecule like an IgG molecule.

**Figure 4 ijms-21-04011-f004:**
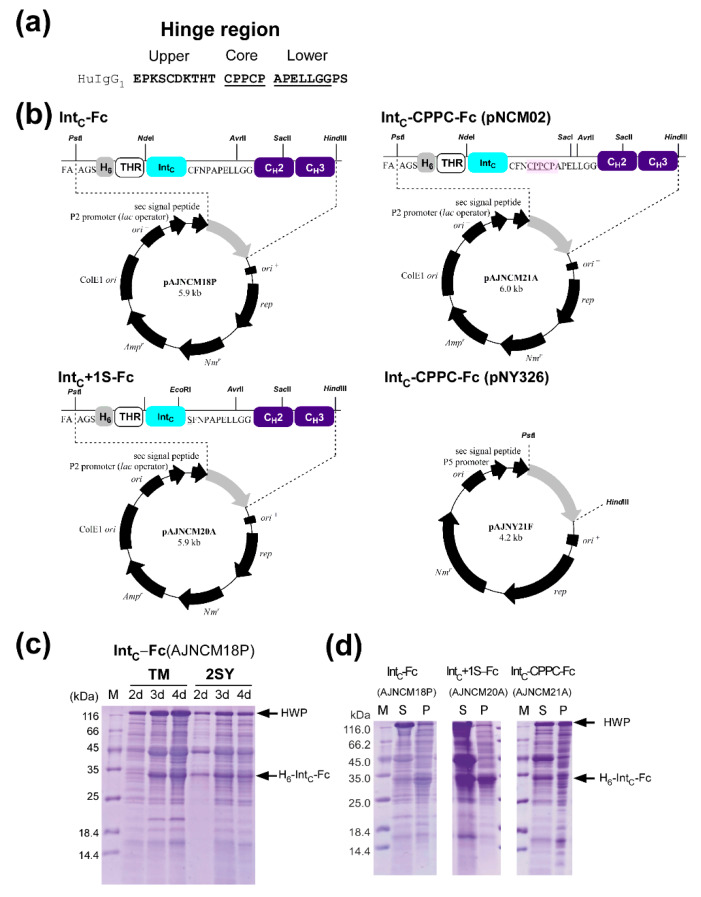
Vector designs and protein production of Int_C_-Fc. (**a**) Primary structure of the hinge region of human IgG. (**b**) Vector maps of three constructs (Int_C_-Fc, Int_C_+1S-Fc, and Int_C_-CPPC-Fc) in the backbone of pNCM02 and pNY326 vector. (**c**) Optimization of the growth media. Production of Int_C_-Fc was compared in TM and SY media. Lanes 2d, 3d, 4d indicate SDS-PAGE analysis of the culture supernatant after 2 days, 3 days, and 4 days of protein expression, respectively. (**d**) Comparison of protein production between Int_C_-Fc, Int_C_+1S-Fc, and Int_C_-CPPC-Fc by SDS-PAGE analysis. Lanes S and P indicate supernatant and pellet fractions from the culture medium after four days of protein expression, respectively. The control cell wall protein (HWP) and Int_C_-Fc are marked with arrows.

**Figure 5 ijms-21-04011-f005:**
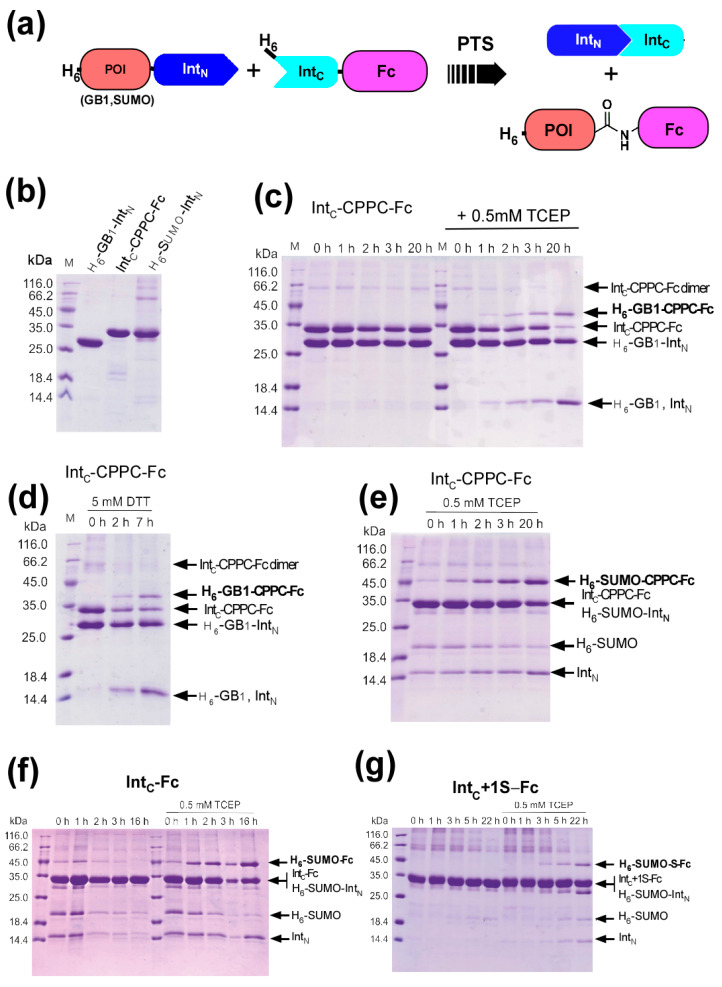
In vitro Fc fusion by PTS (**a**) Schematic drawing of the ligation of model proteins (SUMO and GB1) with the C-terminal Fc domain by PTS. (**b**) SDS-PAGE analysis of the N- and C-terminal precursor proteins (H_6_-GB1-Int_N_, Int_C_-CPPC-Fc, and H_6_-SUMO-Int_N_) independently purified by IMAC. (**c**) Time course of the ligation reaction between H_6_-GB1-Int_N_ and Int_C_-CPPC-Fc in the absence (left panel) or presence (right panel) of 0.5 mM TCEP. Samples were taken after 0, 1, 2, 3, and 20 h of incubation, and analyzed by SDS-PAGE. (**d**) The effect of 5 mM DTT instead of 0.5 mM TCEP was tested for the reaction mixture of H_6_-GB1-Int_N_ and Int_C_-CPPC-Fc. Samples were analyzed by SDS-PAGE after 0, 2, and 7 h of incubation. (**e**–**g**) Comparison between the three Fc constructs (Int_C_-CPPC-Fc (**e**), Int_C_-Fc (**f**), and Int_C_+1S-Fc (**g**)). Protein ligation with H_6_-SUMO-Int_N_ was analyzed with and without 0.5 mM TCEP by SDS-PAGEs. Incubation times in hours (h) are indicated above the lanes of SDS-PAGEs.

**Figure 6 ijms-21-04011-f006:**
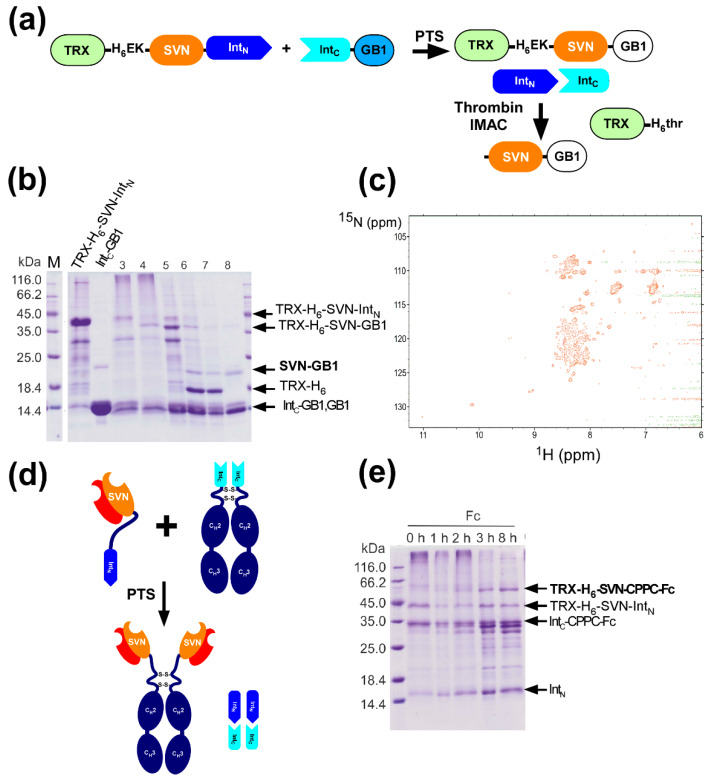
A lectin fusion protein, ‘lectibody’ (**a**) Schematic drawing of the ligation of a lectin (SVN) with a model protein (GB1) by PTS. (**b**) SDS-PAGE analysis of the protein ligation of SVN and GB1. The first two lanes show separately prepared N- and C-precursor proteins (TRX-H_6_-SVN-Int_N_ and Int_C_-GB1). Lanes 3 and 4 show the ligation reaction immediately after the mixture and after 8 h of incubation, respectively. Lane 5 and 6 indicate samples before and after the proteolytic digestion by enterokinase (EK), respectively. Lanes 7 and 8 show samples before loading on a protein A sepharose column and the elution fraction from the column, respectively. (**c**) [^1^H,^15^N]-HSQC spectrum of the segmentally isotopic labeled [^15^N]-SVN-[^14^N]-GB. (**d**) Schematic drawing of the production of SVN-Fc fusion by PTS. (**e**) SDS-PAGE analysis of the time-course of the ligation between TRX-H_6_-SVN-Int_N_ and Int_C_-CPPC-Fc by PTS.

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
