# Peer review of "An Off-the-Shelf Approach for the Production of Fc Fusion Proteins by Protein Trans-Splicing towards Generating a Lectibody In Vitro"

_ijms, 2020, doi:10.3390/ijms21114011_

Round 1
Reviewer 1 Report
1) The term "Y-shape IgG" sounds unusual because, commonly, IgG is Y-shape. It will change to "IgG".
2) The term "lectibody" is recently becoming popular as a cutting-edge technology of protein engineering. Comparing to Hamorsky (ref 21), authors should mention being clear the difference from their achievement.
Primarily, your MS should describe the glycan-binding property and the binding constancy comparing to ref 21 because both lectibodies recognized mannose.
In the case of ref 21, a beta-trefoil folding lectin applied to the engineering. Your MS can compare the structural difference between the two lectibodies as an example of the name of protein foldings.
3) If the MS subjects on lectibody, the description should direct on it more than nobody and anybody in Fig 4. If so, Fig 1 and 2 also improve to be more compact (some of them can move in the part of supplement fig).
4) The mention of the inhibition of the lectibody to COVID-19 in the conclusion should be careful with any evidence to keep the reliability and quality of the journal. From what the evidence, can readers believe it?
Author Response
1) The term "Y-shape IgG" sounds unusual because, commonly, IgG is Y-shape. It will change to "IgG".
Response: We removed "Y-shape” from "Y-shape IgG".
2) The term "lectibody" is recently becoming popular as a cutting-edge technology of protein engineering. Comparing to Hamorsky (ref 21), authors should mention being clear the difference from their achievement.
Response: Pg. 11, line 295-6. We added a sentence to emphasize the differences.
Primarily, your MS should describe the glycan-binding property and the binding constancy comparing to ref 21 because both lectibodies recognized mannose.
Response: In our case, the lectin domain produced was not folded properly. And the lectibody is unlikely to be active. Our MS focuses on the in vitro fusion of Fc by the off-the-shelf approach by protein ligation. One application was a lectin from algae, which was not promising for further characterization as we initially aimed.
In the case of ref 21, a beta-trefoil folding lectin applied to the engineering. Your MS can compare the structural difference between the two lectibodies as an example of the name of protein foldings.
Response: As demonstrated by NMR, the lectin SVN was not properly folded, presumably after the protein ligation. Therefore, the lectibody we produced was unlikely properly folded and, indeed, precipitated during the concentration. We propose other lectins without S-S bridge such as GRFT for the ligation approach in our manuscript.
3) If the MS subjects on lectibody, the description should direct on it more than nobody and anybody in Fig 4. If so, Fig 1 and 2 also improve to be more compact (some of them can move in the part of supplement fig).
Response: We made a new Fig. 1, merging with Fig.2 for making them compact, as suggested.
4) The mention of the inhibition of the lectibody to COVID-19 in the conclusion should be careful with any evidence to keep the reliability and quality of the journal. From what the evidence, can readers believe it?
Response: We thank the reviewer for the comment. We did not claimed that the lectibody could inhibit COVID-19 in our manuscript. We believe that the production of Fc-fusion, including a lectibody, with the off-the-shelf approach by in vitro ligation, could be advantageous over genetic fusions against emerging infectious diseases including COVID-19 because they could be produced with any domains in a few hours without any genetic manipulation. We rephrase the abstract and conclusion to avoid misleading.
Reviewer 2 Report
In this paper, Jaakkonen et al. reported a construction method for Fc fusion proteins using split intein technique. Several intein-C-Fc proteins were expressed by Brevibacillus bacteria with different hinge sequences. Though the construction of lectibody may be unsuccessful because of the instability of a lectin SVN in reducing condition, the paper contains several interesting findings including the successful ligation by serine residue at the hinge of antibody. This reviewer supports the publication of this manuscript after addressing the following points.
- The formation of disulfide bonds between the heavy chains is not confirmed though Fig 7d is depicted there are. In the presence of TCEP, two cysteines at the hinge are reduced. Thus, the formation of the disulfide bonds should be induced by oxidative reagent, such as dehydroascorbic acid and the formation of the bond must be confirmed by a non-reduced SDS-PAGE.
- The citations for the pioneering works for the application of split intein for the off-the-shelf approach to construct bispecific antibody are missing. Han et al. (Sci Rep. 2017,7(1):8360.) and Shibuya et al. (Protein Eng Des Sel. 2017, 30(1):15-21) reported the construction of IgG type and tandem VHH type bispecific antibodies by using split intein. Especially, Han et al. reported the use of IntC-Fc. These reports must be cited.
Minor points:
- Page 8, line 199: Citation is missing with empty parentheses.
- Page 11, line 276: “TRX-H6-CPPC-Fc” should be “TRX-H6-SVN-CPPC-Fc”
- Page 11, line 277: Citation #21 may be cited at a different position.
Author Response
In this paper, Jaakkonen et al. reported a construction method for Fc fusion proteins using split intein technique. Several intein-C-Fc proteins were expressed by Brevibacillus bacteria with different hinge sequences. Though the construction of lectibody may be unsuccessful because of the instability of a lectin SVN in reducing condition, the paper contains several interesting findings including the successful ligation by serine residue at the hinge of antibody. This reviewer supports the publication of this manuscript after addressing the following points.
Response: We appreciate these comments. Indeed, our results showed that SVN was not suitable for this approach and that B. brevisous was not very impressive for the production of Fc-intein fusions due to the difficulty in scaling up on the contrast to the previous reports.
- 1. The formation of disulfide bonds between the heavy chains is not confirmed though Fig 7d is depicted there are. In the presence of TCEP, two cysteines at the hinge are reduced. Thus, the formation of the disulfide bonds should be induced by oxidative reagent, such as dehydroascorbic acid and the formation of the bond must be confirmed by a non-reduced SDS-PAGE.
Response: We added Supplemental Fig. S2. The S-S bridges in the hinge region were formed by oxidation under normal purification and storage conditions unless reducing agents were added. We could confirm the oxidization between two heavy chains by the Cys+1Ser mutation in Supplemental Fig. S3. As all SDS-PAGE analyses were usually performed with 1 mM DTT. We think that hinge S-S bridges of soluble Fc were reduced in SDS-PAGE analysis but formed under non-reducing conditions (Supplemental Fig. S3).
- 2. The citations for the pioneering works for the application of split intein for the off-the-shelf approach to construct bispecific antibody are missing. Han et al. (Sci Rep. 2017,7(1):8360.) and Shibuya et al. (Protein Eng Des Sel. 2017, 30(1):15-21) reported the construction of IgG type and tandem VHH type bispecific antibodies by using split intein. Especially, Han et al. reported the use of IntC-Fc. These reports must be cited.
Response: We thank pointing out these missing references. We now included these two references.
Minor points:
- 1. Page 8, line 199: Citation is missing with empty parentheses.
Response: corrected.
- 2. Page 11, line 276: “TRX-H6-CPPC-Fc” should be “TRX-H6-SVN-CPPC-Fc”
Response: corrected.
- 3. Page 11, line 277: Citation #21 may be cited in a different position.
Response: corrected.
Reviewer 3 Report
Authors present several protein fusions produced by protein-trans splicing, and feature them as a off-the-shelf approach for combinatorial production of Fc fusions in vitro, including lectins as fusion partners. As attractive as this combinatorial “click-in” approach may be, some major issues should be addressed:
- The final product is not isolated from the components of the reaction mixture, and not quantified. This is of essential importance to assess the manufacturability and the developability of the off-the-shelf Fc-fusions. Further, biophysical characteristics need to be demonstrated (at minimum with SEC in native conditions)
- Several aspects of Fc fragment, such as favorable pharmacokinetic properties and ability to elicit effector functions, are mentioned, but none of them demonstrated for the molecules that authors produced. This is essential for bacterially derived proteins. Protein A binding, which authors consider a structural marker, also pertains to some Fab fragments and they demonstrate low micromolar affinity in contrast to 10e-8 typical for Fc. Please rewrite the comments or provide further data.
- There is no demonstration that the developed constructs are able to elicit their binding function – please provide additional data.
- The authors invested a lot of effort to improve the molecular fusion. Evidence that the fusion product is as expected should be given (e.g. mass-spec assisted peptide sequencing).
- As mentioned previously, the functionality of the constructs should be shown in detail (for both “fusion partner” and the Fc fragment, especially after production in reducing conditions.
Please find enclosed also the list of minor remarks that should be addressed. Please afford a careful read-through to improve on vague expressions. Thank you!
Page 1 line 14: antibody fragments
Page 1 line 40: which mediates various effector functions
Page 2 line 44: are therefore termed heavy-chain…
Page 2 line 48: has been widely used as a stand-alone binding agent…
Page 3 line 50 (Figure 1): Different domains and not multi-domains
Page 3 line 51: (B) this is a VHH-Fc fusion
Page 3 line 51 (C): orange and red ovals are both VH – is this typographic, please correct. 2 bispecific antigen-binding entities should be labelled VH1/VL1 and VH2/VL2. If the authors indicate unique chain that is used to produce bispecific Abs by color, the designation of the domains should be corrected.
Page 3 lines 66-68: heterodimeric bispecific IgGs do not share the typical Fc domain with homodimeric antibodies-please reword the sentence.
Page 3 lines 69-72: “genetic engineering” of the cell lines should be replaced with “transfection”
Page 3 line 71: four separate constructs and not genes
Figure 2a: the parts of IgG on the right panel are Fab fragments and the Fc region
Figure 2b: protein trans-splicing
Figure 2b: Immunoglobulin G
Page 4 line 94: extein
Page 4 line 98: IntC (caps)
Page 4 line 104: Gram-positive
Page 4 line 106: and model protein
Page 6 line 120: in vitro in italics
Page 6 lines 130, 132 and throughout the text: Brevibacillus beginning with capital letter
Page 6 line 141: … that bacteria transformed with… vector secreted more potently, … (vector constructs per se do not secrete)
Figure 5b. Please improve the labelling and designation. These are 4 constructs, not 3. PstI site
Page 8 line 173: Staphylococcal with caps
Page 8 line 199: reference missing
Page 11 line 244: In vitro production, in vitro back in normal lettering if the title is in caps
Page 11 line 253: in vivo stability, in vivo in italics
Page 11 line 263: stop missing “After the digestion…”
Page 11 line 266: reducing conditions, not a reducing condition
Page 12 line 305: Brevibacillus in italics
Page 12 line 308: reducing conditions
Page 12 line 324: reducing conditions
Page 12 line 328: long half-life (pharmacokinetic properties) is not an effector function, please reword. Second, how will a bacterially produced Fc devoid of glycosylation induce effector functions? Please explain.
Page 13 line 347: sites, not stis
Page 14 line 392: protein expression
Page 14 line 400: kilodalton is sometimes kD and sometimes kDa, please choose one
Page 14 line 420: slope of the gradient is missing
Author Response
Authors present several protein fusions produced by protein-trans splicing and feature them as an off-the-shelf approach for combinatorial production of Fc fusions in vitro, including lectins as fusion partners. As attractive as this combinatorial “click-in” approach may be, some major issues should be addressed:
The final product is not isolated from the components of the reaction mixture, and not quantified. This is of essential importance to assess the manufacturability and the developability of the off-the-shelf Fc-fusions.
Response: We thank the reviewer for valuable comments. We aimed to produce a large quantity of the Fc-intein fusion using B. brevisous for the manufacturing it for various characterizations as the reviewer suggested because B. brevious has been used for a large quantity of bioactive proteins previously. On the contrary, we could not produce such a large amount (>1g/L) as reported. The yield was translated into 27mg/L, like other previous reports. However, this number was extrapolated from the 4ml culture. And we could not scale up to more than 50 mL. In conclusion, B. brevisous was not suitable for scaling up the production as easily as previous reports. However, the in vitro protein ligation with Fc domain with various binding domains was feasible.
Further, biophysical characteristics need to be demonstrated (at minimum with SEC in native conditions)
Several aspects of Fc fragment, such as favorable pharmacokinetic properties and ability to elicit effector functions, are mentioned, but none of them demonstrated for the molecules that authors produced. This is essential for bacterially derived proteins.
Response: Indeed, it would be ideal to do such experiments. However, the amount we obtained was ca. 1 mg as the starting material of Fc-intein fusion after the first step purification. We added Supplemental Fig. S2 and S4.
Protein A binding, which authors consider a structural marker, also pertains to some Fab fragments and they demonstrate low micromolar affinity in contrast to 10e-8 typical for Fc. Please rewrite the comments or provide further data.
Response: As pointed out, protein A can bind to Fab from VH3 family. However, there was no Fab fragment in the E.coli production because we overexpressed the Fc domain only. The other component from E.coli was removed first by IMAC using His-tag in the intein fragment. We do not believe that Fab could contaminate during the expression in E.coli and purification. In other words, there was no other Protein-A binding protein in any steps of the procedure. Protein A binds the elbow region between CH2 and CH3. Therefore, the Fc domain must fold into a proper conformation for protein A to bind to the Fc domain. We also provided Supplemental Fig. S3 for demonstrating the effects of different reducing conditions on the Fc domain.
There is no demonstration that the developed constructs are able to elicit their binding function – please provide additional data.
Response: As we wrote, we do not think the SVN fusion was functional because NMR spectrum suggested that SVN was not folded properly. However, the protein ligation by PTS was feasible with these domains. We suggest that lectins without disulfide bonds would be more appropriate choices for this approach.
The authors invested a lot of effort to improve the molecular fusion. Evidence that the fusion product is as expected should be given (e.g. mass-spec assisted peptide sequencing).
Response: We now included mass-analysis in Supplemental Figs. S2 and S4. Unfortunately, we could not obtain the molecular mass of the lectibody, presumably because of the aggregation from the unfolded SVN region, and too little amount for collecting the data.
As mentioned previously, the functionality of the constructs should be shown in detail (for both “fusion partner” and the Fc fragment, especially after production in reducing conditions.
Response: Our manuscript focuses on the off-the-shelf approach by in vitro ligation rather than their functionality that we aimed to characterize initially. We aimed at the large quantity production for such an investigation. However, we could not obtain enough quantities for such studies, which we now emphasized in pg. 7 line 174-.
Please find enclosed also the list of minor remarks that should be addressed. Please afford a careful read-through to improve on vague expressions. Thank you!
Page 1 line 14: antibody fragments
Response: we corrected it. Page 1 line 40: which mediates various effector functions
Response: we corrected it.
Page 2 line 44: are therefore termed heavy-chain…
Response: we corrected.
Page 2 line 48: has been widely used as a stand-alone binding agent…
Response: we corrected it.
Page 3 line 50 (Figure 1): Different domains and not multi-domains
Response: we corrected.
Page 3 line 51: (B) this is a VHH-Fc fusion
Response: the variable domain of a heavy chain-only antibodies are often termed VHH such as cameloid antibodies, for example, Bioimpacts. 2013; 3(1): 1–4 doi: 10.5681/bi.2013.009. we corrected the typo.
Page 3 line 51 (C): orange and red ovals are both VH – is this typographic, please correct. 2 bispecific antigen-binding entities should be labelled VH1/VL1 and VH2/VL2. If the authors indicate unique chain that is used to produce bispecific Abs by color, the designation of the domains should be corrected.
Response: We corrected to VH1/VL1 and VH2/VL2
Page 3 lines 66-68: heterodimeric bispecific IgGs do not share the typical Fc domain with homodimeric antibodies-please reword the sentence.
Response: we rephrased.
Page 3 lines 69-72: “genetic engineering” of the cell lines should be replaced with “transfection”
Response: we rephrased.
Page 3 line 71: four separate constructs and not genes
Response: corrected.
Figure 2a: the parts of IgG on the right panel are Fab fragments and the Fc region
Response: Fig 2 is now merged and updated.
Figure 2b: protein trans-splicing
Response: corrected now in Fig.1.
Figure 2b: Immunoglobulin G
Response: removed.
Page 4 line 94: extein
Response: corrected.
Page 4 line 98: IntC (caps)
Response: corrected.
Page 4 line 104: Gram-positive
Response: To our knowledge, it is gram-positive. “Brevibacillus choshinensis HPD52T (DSM 8552) is a Gram-positive, spore-forming, and protein-producing bacterium. Genome Announc. 2016 Jan-Feb; 4(1): e01688-15. doi: 10.1128/genomeA.01688-15
Page 4 line 106: and model protein
Response: Wee rephrased.
Page 6 line 120: in vitro in italics
Response: corrected.
Page 6 lines 130, 132 and throughout the text: Brevibacillus beginning with capital letter
Response: corrected.
Page 6 line 141: … that bacteria transformed with… vector secreted more potently, … (vector constructs per se do not secrete)
Response: we rephrased.
Figure 5b. Please improve the labelling and designation. These are 4 constructs, not 3. PstI site
Response: we improved.
Page 8 line 173: Staphylococcal with caps
Response: corrected
Page 8 line 199: reference missing
Response: corrected
Page 11 line 244: In vitro production, in vitro back in normal lettering if the title is in caps
Response: corrected
Page 11 line 253: in vivo stability, in vivo in italics
Response: corrected
Page 11 line 263: stop missing “After the digestion…”
Response: corrected.
Page 11 line 266: reducing conditions, not a reducing condition
Response: corrected.
Page 12 line 305: Brevibacillus in italics
Response: corrected.
Page 12 line 308: reducing conditions
Response: corrected.
Page 12 line 324: reducing conditions
Response: corrected.
Page 12 line 328: long half-life (pharmacokinetic properties) is not an effector function, please reword. Second, how will a bacterially produced Fc devoid of glycosylation induce effector functions? Please explain.
Response: we rephrased.
Page 13 line 347: sites, not stis
Response: corrected.
Page 14 line 392: protein expression
Response: corrected.
Page 14 line 400: kilodalton is sometimes kD and sometimes kDa, please choose one
Response: corrected.
Page 14 line 420: slope of the gradient is missing
Response: We added about the gradient description.
Round 2
Reviewer 1 Report
"scytovirin" should include in the keyword.
The information on structure and function about the lectin scytovirin is effective to describe in the introduction part with references (44-46) to make readers know the impact of why the article focused the lectin.
The style of references needs to edit. Some texts are all capital words.
Author Response
The information on structure and function about the lectin scytovirin is effective to describe in the introduction part with references (44-46) to make readers know the impact of why the article focused the lectin.
Response: We appreciate the suggestion. we described the function of scytovirin briefly in the introduction with the references.
The style of references needs to edit. Some texts are all capital words.
Response: We updated the style. I believe that some publishers will update from their own database for the reference style consistency.
Reviewer 3 Report
Dear Authors,
thank you for respecting the input, including additional data and making several adaptations to the interpretation of the results, which render the manuscript much more plausible. There was a misunderstanding on my comment on the affinity of Fc to protein A: I was not suggesting that there is a Fab-contamination, but that much lower than 10e-8 affinities can be sufficient for proteinA mediated purification, and solely this property should not be considered sufficient for claiming that the Fc fold has been preserved. Nevertheless, you have reworded the pertaining statements, and I can recommend your manuscript for publication.
Author Response
There was a misunderstanding on my comment on the affinity of Fc to protein A: I was not suggesting that there is a Fab-contamination, but that much lower than 10e-8 affinities can be sufficient for proteinA mediated purification, and solely this property should not be considered sufficient for claiming that the Fc fold has been preserved. Nevertheless, you have reworded the pertaining statements, and I can recommend your manuscript for publication.
Response: As the reviewers pointed out, it could be said that the specificity of protein A is not very high because it could bind to Fab as well. And glycosylated Fc does not have an identical structure as glycosylated Fc, at least according to several crystal structures, even though both could bind to Protein A.
However, we believe that the Fc domain is folded into a globular structure with proper S-S bridges in the core of Ig fold as can be seen from the sensitivities to different reducing conditions.